

# Physical properties of shallow ice cores from Antarctic and sub-Antarctic islands

Elizabeth R. Thomas[1], Guisella Gacitúa[2], Joel B.Pedro[3,4], Amy C.F. King[1], Bradley Markle[5],

Mariusz Potocki[6,7], Dorothea E. Moser[1,8]

[1]British Antarctic Survey, Cambridge, CB3 0ET, UK

[2]Centro de Investigación Gaia Antártica, Universidad de Magallanes, Punta Arenas, Chile

[3]Australian Antarctic Division, Kingston, 7050, Australia

[4]Australian Antarctic Programme Partnership, Hobart, Tasmania 7001 Australia

[5]California Institute of Technology, Pasadena, CA, USA, 91125

[6]Climate     Change     Institute,     University     of     Maine,     Orono,     ME     04469,     USA

[7]School of Earth and Climate Sciences, University of Maine, Orono, ME 04469, USA

[8]Institut für Geologie und Paläontologie, University of Münster, 48149 Münster, Germany

**Abstract.**

The sub-Antarctic is one of the most data sparse regions on earth. A number of glaciated Antarctic and sub-Antarctic islands have the potential to provide unique ice core records of past climate, atmospheric circulation and sea ice. However, very little is known about the glaciology of these remote islands or their vulnerability to
warming atmospheric temperatures. Here we present ground penetrating radar (GPR), melt histories and density profiles from shallow ice cores (14 to 24 m) drilled on three sub-Antarctic islands and two Antarctic coastal domes. This includes the first ever ice cores from Bouvet Island (54°26'0 S, 3°25'0 E) in the South Atlantic, from Peter 1[st] Island (68°50'0 S, 90°35'0 W) in the Bellingshausen Sea and from Young Island (66°17'S,
162°25'E) in the Ross Sea sector's Balleny Islands chain. Despite their sub-Antarctic location, surface melt is low at most sites (melt layers account for ~10% of total core), with undisturbed ice layers in the upper ~40 m, suggesting minimal impact of melt water percolation. The exception is Young Island, where melt layers account for 47% of the ice core. Surface snow densities range from 0.47 to 0.52 kg m³, with close-off depths ranging from 21 to 51 m. Based on the measured density, we estimate that the bottom ages of a 100 m ice core drilled on
Peter 1[st] Island would reach ~1836 AD and ~1743 AD at Young Island.

## 1 Introduction

The sub-Antarctic region sits at the interface of polar and mid-latitude climate regimes, making it highly sensitive to shifting climate over time. However, the sub-Antarctic and the Southern Ocean is one of the most
data sparse regions on earth. A recent compilation of climate data spanning the 20[th] century revealed that of the 692 records that exist globally (Emile-Geay et al., 2017), none are available for the Southern Ocean. The vast expanse of open water makes climate and paleoclimate observations difficult, but a number of glaciated islands



from this region may be suitable for extracting paleoclimate information from ice cores (Figure 1). Given their location, these sub-Antarctic islands (SAIs- defined here as islands south of the Southern Ocean polar front) could provide valuable archives of southern hemisphere westerly winds and sea ice variability, key components of the global climate system.

Westerly winds drive ocean upwelling and basal melt of Antarctic ice shelves (Favier et al., 2014; Joughin et al., 2014) and the 20[th] century increase in Antarctic snowfall has been attributed to changes in their circulation (Medley and Thomas, 2019). Sea ice modulates the earth's albedo and governs the area available for exchange of heat and $CO_2$ between the ocean and the atmosphere.  However, records of both westerly winds and sea ice are short, with major uncertainty in the trends prior to the satellite era (i.e. pre 1970's).

Many of these SAIs sit within the westerly wind belt and the transitional sea-ice zone (Figure 1). Those on the very margin of maximum winter sea-ice extent could potentially identify local-scale changes in sea ice, rather than relying on ice core records from the Antarctic continent, which record mostly ocean-sector scale trends (Thomas and Abram, 2016; Thomas et al., 2019). An ice core from Bouvet Island, presented in this study, has already proven successful in reconstructing past sea ice variability in the adjacent ocean (King et al., 2019).

Paleoclimate records do exist on some SAIs from lake sediments, peat cores and ice cores. Peat records are currently the most prolific of the three, with a number of records retrieved on the New Zealand SAIs up to 12,000 years old (McGlone, 2002) and on Crozet and South Georgia up to 6000 and 11,000 years old respectively (Van der Putten et al., 2012). One limitation, is that annual layers are not preserved in such records and carbon dating carries with it uncertainties in the decadal to centennial range, which complicates the identification of annual to decadal-scale climate variations and the precise timing of climate shifts (Van der Putten et al., 2012). Peat deposits are also lacking on the more southerly SAIs.

Lake sediments are similarly useful for studying vegetation composition on the SAIs. These are related to a wide range of environmental parameters within and around the lake used to reconstruct past natural variability (Saunders et al., 2008). A novel proxy, based on sea-salt aerosols in lake sediments on Macquarie Island, reconstructed past westerly wind strength over the past 12,300 years (Saunders et al., 2018). However, damage from introduced rabbits at this site has compromised the data during the last ~100 years.

Ice core records offer complementary archives to peat and lake archives on glaciated or ice capped islands and are in some cases unique records where peat and lake deposits are not available. Shallow ice cores have been drilled from low elevation plateaus and glacial terminus on South Georgia, the largest glaciated SAI, revealing ice ages of between 8000-12000 years old (Mayewski et al., 2016). However, surface temperatures on South Georgia have risen by ~1°C during the 20[th] century (Whitehead et al., 2008) and ninety-seven percent of the 103 coastal glaciers have retreated since the 1950's (Cook et al., 2010). Thus, the valuable paleoclimate archive contained in South Georgia, and potentially the other SAIs glaciers, may be at risk from surface melt.



The sub-Antarctic Ice Core drilling Expedition (subICE) was part of the international Antarctic Circumnavigation Expedition (ACE) 2017-2018 (Walton, 2018), which offered an exceptional opportunity to access multiple SAIs. In this study we present the density profiles, melt histories and ground penetrating radar data collected during subICE. These include the first ever ice core records from three of the glaciated Antarctic and SAIs in the Pacific and South Atlantic sector of the Southern Ocean, together with two continental ice cores collected from coastal domes in East and West Antarctica. The aim of this study is to 1) evaluate the ice conditions and internal layering in the upper ~40 m, 2) determine the extent of surface melt and 3) estimate potential bottom ages of future deep drilling expeditions.

## 2 Methods and Data

### 2.1 Meteorological data

Meteorological data come from the European Centre for Medium-Range Weather Forecasts ERA-5 analysis (1979–2017) (Copernicus Climate Change Service, 2017), the fifth generation of ECMWF reanalysis. ERA-5 reanalysis currently extends back to 1979, providing hourly data at 0.25-degree resolution (~ 31 km). We note that the resolution of ERA-5 is currently unable to capture local climate conditions on the SAIs however, a recent study from the Antarctic Peninsula confirmed its high accuracy in representing the magnitude and variability of near-surface air temperature and wind regimes (Tetzner et al., 2019).

Limited in-situ meteorological observations are available from the University of Wisconsin-Madison Antarctic Meteorology Program http://amrc-new.ssec.wisc.edu/aboutus/. Automatic weather station (AWS) data is available from Peter 1st Island,(September 2006 to January 2007), Mt Siple (January 1992 to January 2015) and Young Island (January 1991 to December 1997). Meteorological observations from Bouvet are available from the Norwegian Polar Institute. https://doi.org/10.21334/npolar.2014.ba8d71fc. WMO Station 689920, 42 m above sea level (April 1997-December 2005).

The 2 m temperatures from ERA-5 (Figure 1) and the AWS sites (typically at low elevations) are expected to be warmer than the ice core drilling locations, as a result of the adiabatic rate of temperature change for vertically moving air. This lapse rate varies with both temperature and mixing ratio from a dry rate of 0.98 deg/100 m to a saturated rate of 0.44 deg/100 m at 20°C. As a best guess we use a lapse rate of 0.68 deg/100 m to calculate the temperature at the drilling sites. This value is observed on the western Antarctic Peninsula (Martin and Peel., 1978), where the climate and maritime conditions are expected to closely resemble those of the sub-Antarctic islands. The altitude corrected value is presented in the text and compared with the 2 m temperature and uncorrected AWS temperature in Table 1.

### 2.2 Ice cores

Seven ice cores were drilled as part of subICE, between January and March 2017 (Table. 1). The teams were deployed to the ice core sites via helicopter from the ship (Akademik Tryoshnikov), allowing between three and eight hours of drilling time, dependent on weather conditions and logistical capability. The ice cores were drilled using a Mark III Kovacs hand-auger with sidewinder winch and power-head, retrieving ice sections of approximately 70 cm in length. The ice was measured, placed inside pre-cut layflat tubing and packed in



insulated ice core storage boxes for transportation. During the voyage, the ice was stored in a -25°C freezer and
later transported to the ice core laboratories at the British Antarctic Survey (BAS).

### 2.2.1 Adelie Land Coast (Mertz)

Two ice cores were drilled on the coast of Adélie Land in East Antarctica (Figure 2a); a 20 m core (Mertz 1)
from Cape Hurley, a low elevation ice dome on the eastern side of the Mertz Glacier (67° 33' S, 145° 18' E); and
a 9 m ice core (Mertz fast ice) drilled on a triangular wedge of fast ice in Fisher Bay (67° 26' S, 145° 34' E),
bounded on the east by the Mertz Glacier and on the west by the AAE (Australian Antarctic Expedition) glacier.
The annual average temperature at this location (ERA-5 elevation corrected) is -17.4°C, making it our coldest
site.

The Mertz Glacier extends into the ocean from coastal King George V land, with a floating ice tongue. The
tongue traps pack ice upstream forming the Mertz Glacier Polynya along its western flank during winter, the
third most productive polynya in Antarctica, (Lacarra et al., 2014). In 2010, the impact from the B9B iceberg
caused this tongue to calve off producing a ~80 km long iceberg. This event had a profound impact on local sea
ice conditions and the formation of dense shelf water (Campagne et al., 2015).

### 2.2.2 Young Island

A 17 m ice core was drilled on Young Island (66°17′S, 162°25′E), the northernmost island in the Balleny Island
chain (245 km²), off the coast of Adélie Land (Figure 2b). The Balleny Islands comprise three major dormant
volcanic islands, Young, Buckle and Sturge (Hatherton et al., 1965), which sit in the Antarctic seasonal sea ice
zone at the boundary of the polar westerlies and Antarctic coastal easterlies. Young Island is characterized by a
thick ice cover, marine-terminating piedmont-glacier tongues, steep coastal cliffs, and is therefore described as
"among the most inaccessible places in the world" (Hatherton et al., 1965). The core was drilled 238 m above
sea level.

The annual average temperature from this site is -9.2°C, from ERA-5 (1979-2017), with a maximum
temperature of 2.8°C recorded from an AWS deployed at 30 m above sea level (1991-1997). On a seasonal
scale, the AWS reveals average air temperatures fluctuate between -1.7°C in summer (November–February) and
-13.9°C during winter (May–August).

### 2.2.3 Mount Siple

A 24 m ice core was drilled from Mount Siple (73°43'S, 126°66'W) on the Amundsen Sea coast, West
Antarctica (Figure 3a). Mount Siple is an active shield-volcano rising to 3,110m at its peak from Siple Island on
the coast of Marie Byrd Land, surrounded by the Getz ice shelf. The core was drilled at 685 m above sea level.
The annual average temperatures is -9.6°C (ERA-5, 1979-2017), with average summer temperatures close to
zero (-0.6°C). However, an AWS deployed near this site (230 m a.s.l, 1992-2015) suggests a maximum
temperature of 2.4°C (elevation corrected).

### 2.2.4 Peter 1st Island





A 12 m ice core was drilled from Peter 1st Island (68°50'0 S, 90°35'0 W), in the Bellingshausen Sea (Figure 3b). The former shield volcano (154 km²) is almost completely covered by a heavily crevassed ice cap and sits within the seasonal sea ice zone. The core was drilled on a ridge (Midtryggen) at 730 m above sea level, in a

small saddle on the eastern side of the island overlooking the main glacier Storfallet. The annual average temperature at this site is -9.5°C (ERA-5, 1979-2017), with summer maximum below zero (-2.7°C).

### 2.2.5 South Georgia

Two ice cores were drilled on South Georgia (54°17'0 S, 36°30'0 W), the largest SAI (3755 km²). A 2.2 m ice

core from the Nordenskjold Glacier, Cumberland Bay, and a 1.8 m core from Heany Glacier in St Andrews Bay (Figure 4a), both on the eastern coast of the island. South Georgia was the warmest island visited, with annual average temperatures of 1.6°C recorded from a near-continuous AWS record from Grytviken, located in Cumberland bay (1905-present). The average summer temperature exceeds 5°C, with a maximum temperature of 8.4°C recorded in February 1907.


The South Georgia ice cores are from glacial terminus sites and do not provide contemporary climate information. Drilling at this site was difficult due to the high temperatures, with evidence of water in the borehole. These cores are therefore not included in this study, but they will provide estimated bottom ages and an evaluation of proxy preservation required for future drilling campaigns at higher-altitude sites (future study).


### 2.2.6 Bouvet Island (Bouvetøya)

A 14 m ice core was drilled from the volcanic island of Bouvet in the central South Atlantic (54°26'0 S, 3°25'0 E, 50 km²), also known as Bouvetøya (Figure 4b). This was the most northerly site but the islands location within the polar front (Figure 1) classifies it as sub-Antarctic. Despite its northerly location the average yearly

temperature from ERA-5 (elevation corrected) is -2.9 °C, with a maximum monthly temperature of -0.9°C. The maximum value recorded by the AWS at 42 m above sea level was 6.7 °C (1997-2005, elevation corrected).

Bouvet is almost entirely ice covered (~50 km²) with the exception of a few rocky outcrops around the coast. Ice flows down the flanks of the volcano, with no visible crevassing, giving way to shear ice-cliffs and near-vertical

icefalls into the sea. The island is the southernmost extension of the mid-Atlantic ridge, located at the triple junction between the African, South American and Antarctic plates. These pronounced sea floor features drive the cold Antarctic Circumpolar Current close to the island, keeping surface temperature cold and allowing sea ice to extent north of the island.

The last known volcanic eruption on Bouvet was 50 BC; however, visible ash and dust layers suggest eruptions may be more frequent. At a number of locations, the ice edge has broken vertically away (Figure 4c) revealing horizontal bands of clean and dirty layers in the vertical stratigraphy. The islands remote location, and absence of significant local dust sources, suggests a local volcanic source from either Bouvet or the South Sandwich Islands. The 3.5 km wide Wilhelmplataet caldera appears entirely ice filled. This potentially offers the deepest

coring location, however it is unclear if this is instead an ice bridge formed since the last eruption.



### 2.3 Ground Penetrating Radar

Ground Penetrating Radar (GPR) measurements were performed around each ice core site. We used a SIR3000 unit equipped with a 400 MHz central frequency antennae (GSSI Inc.). The system was pulled on a sledge while walking in parallel and transversal lines of 100 m to 500 m depending on the site surface conditions and available time at each site. The penetration depth of the propagated wave is strongly controlled by the electrical properties of the subsurface combined with the central frequency of the system. The equipment used provides a good compensation between these parameters in polar snow. Vertical resolution is approximately 0.35 m, reaching up to 100 m depth in very dry snow (ideal conditions), such as in the Antarctic plateau (Spikes et al., 2004). However, the resolution and maximum reachable depth are expected to decrease dramatically at lower latitudes. This GPR was intended to obtain data from the near surface to complement the ice-core observations and to better characterise the site spatially.

Data collected is observed in-situ as a 'radargram' that represents the number of traces received (x-axis) and the two-way travel time of the wave in nano seconds (ns) (y-axis), which is the time the signal takes from the transmitter to the receiver when reflected from the internal ice discontinuities. During data collection the maximum time window was set to a value between 400 – 600 ns, according to the expected maximum depth of signal propagation at the SAI sites. GPR data was monitored in-situ for calibration and stored. Data processing was done using a commercial software (ReflexW) and generally included: removal of repetitive traces (same position), correction of the surface position and distance covered, frequency filters, gain function adjustments, stacking and other visual enhancements to improve the interpretation of the reflecting layers. Each collected file was analysed independently, and layers were picked manually in full resolution. Thus, if layers were not sufficiently clear and continuous, they were not picked.

In order to obtain the corresponding depth (m) for the y-axis, we used the density profile (see section 3.3) of the ice core to obtain the average velocity of the wave (m/ns) in the ice based on the Looyenga model (Looyenga, 1965) for each site.

### 2.4 Ice core analysis

Ice-core processing was carried out in the -20°C cold laboratories at BAS. The section length, diameter and weight were measured to provide a density record and the visible melt layers logged and measured (only layers > 1mm). Melt layer thickness was corrected for ice thinning based on the Nye model (Nye, 1963), that assumes a vertical strain rate and thinning that is proportional to burial.

## 3 Results and discussion

### 3.1 GPR

Grids of parallel and transversal lines traced for all sites are summarized in Table 2. Given the scope of this paper, one example of a representative profile taken in the area nearby the ice core borehole is shown in the following section.

### 3.1.1 Adelie Land Coast (Mertz)



The Mertz 1 site (Cape Hurley) had a flat surface, consistent with observed internal layering. Layers are not distinguished in the upper ~7 m of snow, below this depth reflectors were not strong, however eleven distinct layers were identified, some of which were discontinuous. Layers were visible down to a depth of 62 m, which we estimated to be the approximate limit of signal propagation of the GPR system at this site. Figure 5 shows a

section of a profile taken in north westerly direction. Bedrock was not detected.

Data quality for the Mertz fast ice site was very poor (Figure 6). A single continuous layer at 6.8 m depth was the only one identified. This site is fundamentally different in character from the others presented in this study. While we believe the surface snow to be meteoric, it does not sit atop grounded ice, but rather a large wedge of multiyear sea ice, held fast between the AAE and Mertz Glacier tongues (Figure 4a). The top-most layers of

snow and firn appeared typical. However, at a drill depth of 6.23 meters the recovered ice was saturated with liquid seawater (a strong attenuator of the radar signal). Ice recovered below this depth was different in character from the surface, being solid ice containing bubbles and interstitially saturated with saltwater. There was a standing water table in the borehole at this same depth. These observations, together with the clear radar reflector at just over 6.8 m (Figure 6), the absence of reflectors below this, and the sites elevation of ~6 m above

sea level, lead to the conclusion that the fast ice is saturated with seawater below sea level.

If the fast ice is floating (i.e. supported buoyantly), our measurement of the dry freeboard thickness allows us to estimate the total thickness of the fast ice wedge. The average density of the top 6 m is 0.55 +/- 0.05 kg m$^{-3}$, increasing to 0.85 ± 0.05 kg m$^{-3}$ below this. Assuming a seawater density of 1.0275 ± 0.004 kg m$^{-3}$, freeboard

thickness of 6.2 m and a mean density of the ice below the water-table of 0.85 +/- 0.05 kg m$^{-3}$, the total ice-equivalent thickness of the ice at this site is ~30 m. However, there are large uncertainties in this calculation. The radargram indicates that the fast ice increases in thickness away from the drill site, by several meters. While this site is not likely to provide typical geochemical proxy records, it may be of interest to future studies of multiyear sea ice.

### 3.1.2 Young Island


The Young Island site was flat with a compact snow surface. Multiple layers can be identified but they are not traceable through the full profile length. The small distance between layers resulted in merging. As an example, on Figure 7, one strong layer (~4 m depth) has been traced along the profile. The maximum estimated depth of detected layers was ~36 m (Table 2) and bedrock was not detected.


### 3.1.3 Mount Siple

The surface of the studied area at Mount Siple site was relatively smooth (max. 4° slope). Figure 8 shows a full profile taken in an east-westerly direction rising from 678 m a.s.l. to 685 m a.s.l. Multiple layers are clearly interpreted with minor discontinuities, for the full depth of the profile (~36 m). Bedrock was not detected.


### 3.1.4 Peter 1st Island

The surface of the Peter 1$^{st}$ island site was smooth (maximum 5° slope) increasing from 718 m a.s.l. up to 726 m a.s.l. in a north-westerly direction. Layers are only (and partially) traceable for the upper ~25 m of the snow/firn





pack. The maximum time window was set to reach an estimated depth of ~43 m (Table 2) and bedrock was not
detected. Figure 9 shows a section of a profile crossing the ice core position.

### 3.1.5 Bouvet Island

Figure 10 shows a radargram obtained at Bouvet descending from 357 m a.s.l. to 339 m a.s.l towards the south
east. The visible layers show that there is a maximum of 2 m of horizontal continuous layering of snow. Below
this layer, there is stratified firn that has been eroded by the effect of wind and the surface slope, reaching a
layer of solid ice between 12 and 18 m from the surface. Another layer is detected between 35-41m depth,
which is interpreted as the bedrock. This depth is consistent with the estimated ice thickness observed from the
ice cliffs to the south of the ice core drilling site (Figure 4c). Ice thickness from the photograph is estimated
relative to the expected size of the observed fur seals on the beach.


### 3.2 Melt records

Surface melting occurs in response to a positive energy balance at the snow surface and is strongly correlated
with surface air temperature. The Antarctic ice sheet experiences little melt, due to consistently low
temperatures, however the coastal margins and areas of the Antarctic Peninsula are subject to surface melting
(van Wessem et al., 2016). The relationship between surface melt and surface temperatures has been exploited
to reconstruct past climate in Antarctic ice cores (Abram et al., 2013). However, the presence of too much
surface melt can damage the climate proxies they contain.

The influence of melt-water on ice core proxy preservation has been explored for arctic ice cores (Koerner,
1997). Here, melt-water percolation can allow insoluble micro particles to migrate to the melting surface,
causing a spike in concentrations, and influence the stable water isotope record, a commonly used proxy for past
surface temperatures. Seasonal melting can cause run-off of near-surface snow and melt and refreezing at the
base of the annual snowpack. This can redistribute the stable water isotope profile, resulting in a warm or cold-
biased record. The influence on both stable water isotopes and chemistry can have a major impact on the ability
to date ice cores using annual layer counting.

Given their location, we expect all our sites to experience some degree of melt. Surprisingly, the site least
effected by melt is Bouvet, the warmest and most northerly location. The average melt layer thickness in the
Bouvet core is 0.3 cm, observed at a frequency of five layers per meter; with the largest measured melt layer just
3.98 cm (Table 2). The average melt layer thickness at all sites (except Young Island) is considerably lower than
the mean melt layer width of 3.2 cm observed at the James Ross Island ice core, from the northern Antarctic
Peninsula (Abram et al., 2013). The percentage of melt per year is dependent on the measured snow
accumulation. The estimated snow accumulation at Bouvet is 0.59 m water equivalent, based on annual layer
counting of chemical and isotopic species (King et al., 2019b). The annual layer counting of the other SUBICE
cores has not yet been completed, however based on our estimates (section 3.4), it is a reasonable assumption
that the snow accumulation at these maritime islands will be equal to or more than the snow accumulation at
James Ross Island (0.62 +/- 0.14 m water equivalent). Thus, the potential for proxy preservation is promising.



The site most affected by melt is Young Island, which has frequent melt layers averaging 6.57 cm and the largest single layer of 61 cm (58 cm before thinning applied). Young Island sits within the Polar Front and the

sea ice minimum (Figure 1), with average temperatures 6.4 degrees colder than Bouvet. The average summer temperature (December- February) at Young Island is -2.18°C. The maximum recorded temperature from an Automatic Weather Station (AWS) deployed at 30 m elevation was 4.2°C (1991–1997), equivalent to 2.8°C at the ice core site. It is very likely that the resolution of ERA-5 is not sufficient to capture local surface temperatures, however, given the islands location south of the Polar Front and the seasonal sea ice zone it seems

unlikely that surface temperature alone can explain the observed melt layers.

### 3.3 Density

The change in firn density with depth is dependent on the snow accumulation rate and temperature at the site. Higher temperatures and lower accumulation rates result in the greatest change of density with depth. At our sites the depth-density profile exhibits more variability at the island sites (Bouvet, Peter 1$^{st}$, Young) than the

continental sites (Mertz and Mt Siple), reflecting the warmer island locations and the influence of surface melt (Figure 11).

A fitted density curve is applied based on the assumption that firn densification is linearly related to the weight of overlying snow (Herron and Langway, 1980). The density is calculated following Eq. (1):


$ph=p_i\text{-exp (ln a +b)}$                                                          (1)

where $p$ is the density, $p_i$ is the density of ice (0.917 kg m$^{-3}$) and $h$ is the snow depth. The constants a and b are derived from the linear relationship between ln $[p/(p_i\text{-}p)]$ and depth, where ln(a) is the intercept, and b is the

gradient.

The first stage of densification relates to grain settling and packing and occurs below the "critical density" of about 0.55 kg m$^{-3}$ (Herron and Langway, 1980). A linear relationship exists between the 2-m temperature from ERA-5 and the critical depth (r$^2$ = 0.57, p>0.05), which is reached first at Bouvet, the most northerly location and the warmest site (Table 1; Table 2).


The second stage of densification (0.55–0.83 kg m$^{-3}$), occurs when the firn air passages become closed off to form individual bubbles. The density at the bottom depth of all cores remains below this value, with the exception of Young Island, which exceeds this limit at 16.6 m. This suggests that pore close off has been achieved below this depth, when air can no longer be excluded and further densification takes place by

compression of the bubbles. However, the presence of a large melt layer at this depth suggests this may be the cause of the density increase. Based on the fitted density profile the actual close-off depth occurs at 21 m. The second fastest close-off depth occurs at Bouvet (28.65 m), considerably shallower than the two coastal Antarctic sites (Mt Siple and Mertz 1).



Surface densities, critical depths and close-off depths at the continental sites (Mt Siple and Mertz 1) are consistent with modelled values (van den Broeke, 2008). A compilation of observed (and modelled) Antarctic snow densities suggest a range of 6- 26 m (modelled 4-29.5 m) for the critical depth and 34-115 m (modelled 45-115 m) for pore close-off. The values for Bouvet, Peter 1[st] and Young Island are close to the modelled values for coastal Antarctica, however, the pore close-off depth at both Bouvet and Young Island is achieved faster
than the lowest reported depth for an Antarctic site (34 m at Upstream B, West Antarctica)(van den Broeke, 2008). Although the presence of melt layers may be affecting out estimate of the close-off depth. In addition to the influence of temperature, the rapid densification on the islands may be caused by layer stretching and compression related to ice flow that is not well understood at these locations.

### 3.4 Estimating the ice core bottom ages

Based on the fitted density curve and assuming a constant rate of snow accumulation, we can estimate the expected bottom ages of the ice cores drilled at each of our sites. We estimate the (water-equivalent) accumulation rate at the annual average precipitation minus evaporation (P-E) value from ERA-5. We note that the resolution of ERA-5 may not be adequate to capture P-E at these sites and that snow accumulation may vary considerably at these island locations. However, the annual average P-E value for Bouvet, the only site that has
been annual layer counted, is identical to the calculated snow accumulation in meters water equivalent.

The estimated ice core bottom ages range from 2001 (+/-2 years), at both the Peter 1[st] and Bouvet ice cores, to 1992 (+/- 3 years) at the Mertz 1 (Table 3). It is difficult to estimate a potential age limit for a core drilled to bedrock at the Young, Peter 1st and Mertz 1 sites since the ice thickness was beyond the penetration depth of the
GPR. For example at Mertz 1 a bottom age of 1919 AD (+/- 5 years) is estimated based on a GPR bottom depth of 62 m (Table 3), although the ice could feasibly be much deeper than this. In Table 5 we estimate the potential ice ages at the signal penetration depth of the GPR at Young, Peter 1[st] and Mertz as 1967, 1948 and 1919 AD respectively. At the Bouvet site, where we interpret the layer at 41 m as bedrock, the maximum potential age is 1962 AD (+/- 5 years). However, considerably deeper (and therefore older) ice may be present at higher
elevations. These results suggest that the SUBICE ice-core records will be suitable for obtaining sub-annual resolution climate records with the potential to capture climate variability over multi-decadal timescales.

For the sites where bedrock was not detected, we estimate the bottom age of an ice core drilled to100 m depth (Table 3). This suggests that the oldest ice would be reached at Mertz 1 (1742 +/- 10 years) and the youngest ice
at Peter 1[st] (1836 +/- 10 years).

The estimated ice core bottom age at Bouvet Island, together with the observed visible ash layers (Figure 4c), offer a glimpse at the volcanic history of this remote island. The last known eruption was 50 BCE; however, visible layers in the upper ~20 m suggest the island has been volcanically active as recently as ~ 2012. The
regularity of the layers in the ice cliffs, visible all around the island, indicate frequent volcanic activity has occurred during the 20[th] century. This is consistent with the persistent volcanic activity observed on the near-by South Sandwich Islands, with visible ash clouds identified in the satellite records and even visible from the ship during the ACE (March 2017). We note that the visible ash layers at Bouvet may have been deposited from the South Sandwich Islands but further analysis will help establish the source.





**4 Conclusions**

Initial results from five shallow ice cores drilled in the sub-Antarctic islands and coastal Antarctica suggest that these locations may be suitable for short-term climate reconstructions and up to centennial-scale reconstructions at some sites if deeper cores could be retrieved. Evidence that these ice cores span the 20[th] century, a period of significant global climate change, is exciting. The GPR surveys suggest relatively uniform layering at most

sites, at least to the depth of the ice cores, suitable for ice core proxy reconstructions. However, evidence of crevassing at some locations (Young and Peter 1[st]) demonstrates the importance of a thorough geophysical survey before contemplating deeper drilling at these sites. Evidence from Bouvet Island reveals regular volcanic ash deposits during the 20[th] century, suggesting the island is still volcanically active.

The impact of melt is less severe than expected at some locations, especially Bouvet Island, but more severe at others. Young Island, part of the Balleny Island chain off the coast of Adèlie Land, is the most susceptible to melt. However, the observed melt layer thickness at the other sites is less than that observed from the James Ross Island ice core, which yielded paleoclimate reconstructions (Abram et al., 2013). Proxy preservation was not a concern in this ice core, suggesting that melt will also not adversely influence the climate record contained

in the sub-Antarctic ice cores. The observed increase in melt intensity at James Ross Island since the mid-20[th] century was linked to warming surface temperatures. Thus the comparable melt intensity observed at some of the SAIs may be evidence that the 20[th] century warming at these locations was analogous to that on the Antarctic Peninsula.

Based on the measured density profile and the P-E from ERA-5, the estimated bottom ages of the SUBICE cores range from 2001 (Peter 1[st] and Bouvet) to 1992 (Mertz 1), suggesting that these records should contain a multi-decadal record of climate variability in this data sparse region. We were unable to obtain ice thickness estimate for all sites, with the exception of Bouvet, however visible layers were identified in the GPR records to depths of 60 m. Even with a conservative estimate, it is possible that deeper ice core drilled on these SAIs would have

the potential to capture climate variability during the 20[th] century, but most likely considerably longer.

**Data availability**

All data will be stored at the UK Polar Data Centre (https://www.bas.ac.uk/data/uk-pdc/) or by directly contacting Liz Thomas (lith@bas.ac.uk). DOI to be provided following paper acceptance.


**Author contributions**

ERT lead the project; ERT, GG, JP, ACFK, BM, and MP collected the data in the field; ERT, JP, ACFK and DEM processed the ice core data; GG and MP processed the GPR data; all authors contributed to writing and editing the paper.


**Competing interests**

All authors declare no competing interests.

**Acknowledgements**



SUBICE received funding from École Polytechnique Fédérale de Lausanne, the Swiss Polar Institute, and
Ferring Pharmaceuticals Inc. ERT received core funding from the Natural Environment Research Council to the
British Antarctic Survey's Ice Dynamics and Paleoclimate Program. AK was jointly supported by Selwyn
College, Cambridge, and the NERC Doctoral Training Programme (Grant NE/L002507/1). JBP acknowledges
support from the European Research Council under the European Community's Seventh Framework Programme
(FP7/2007e2013)/ERC Grant Agreement 610055 as part of the ice2ice project. We are grateful to the
Norwegian Polar Institute for granting us permission to visit Bouvet (permit ref: 2016/115-25). The authors
appreciate the support of the University of Wisconsin-Madison Automatic Weather Station Program for the data
set, data display, and information, NSF grant number ANT-1543305 and ECMWF for providing ERA-5
reanalysis data. We thank Laura Gerish (BAS) for producing the maps. Data used in this study are available
through on the UK Polar Data Centre. The authors would like to acknowledge the coordinators and participants
of the Antarctic Circumnavigation Expedition for facilitating collection of the subICE cores, especially David
Walton, Christian de Marliave, Julia Schmale, Robert Brett, Sergio Rodrigues, Francois Bernard, Roger Stilwell
and Frederick Paulsen.

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

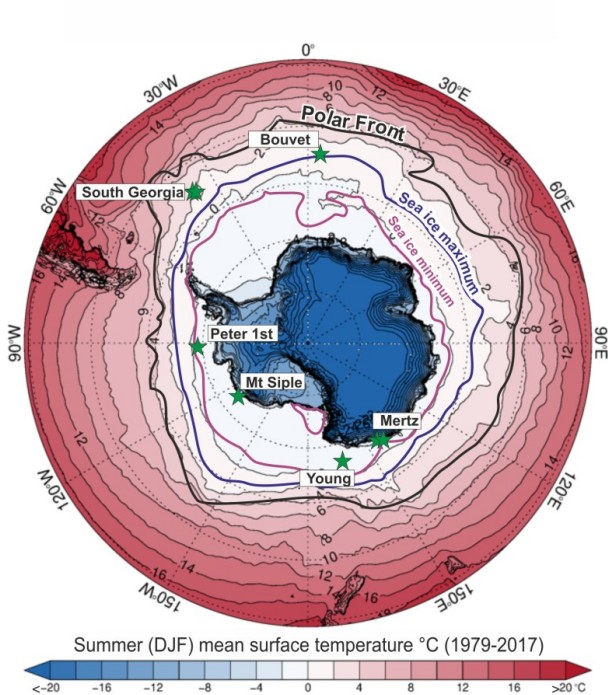

**Figure 1: Map of subICE ice core locations (stars). Overlain on mean summer surface temperatures (coloured contours) from ERA-5 (1979-2017). Location of the maximum (blue) and minimum (purple) sea ice extent, from NSIDC (1981-2017), and the Polar Front (black).**


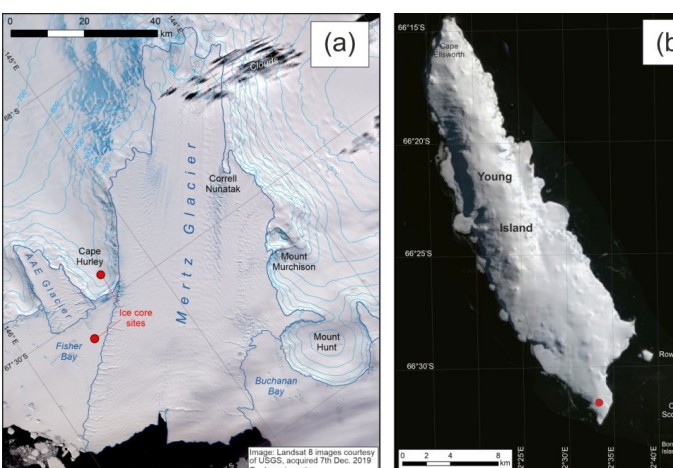

**Figure 2: Indian sector ice cores. (a) Map of Mertz glacier, showing the location of the two ice cores from Cape Hurley and Fisher Bay. (b) Map of Young island, showing ice core location (red circles). Landsat image courtesy of USGS. Contours derived from TanDEM-X 90m DEM**


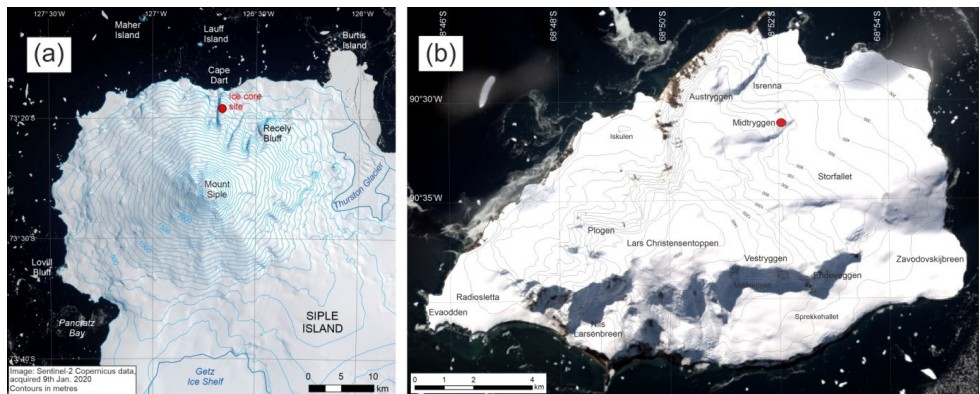

**Figure 3: Pacific sector ice cores. Map of (a) Mount Siple and (b) Peter 1st island showing locations of ice cores (red circles). Image from Sentinel-2 Copernicus data. Contours derived from TanDEM-X 90m DEM**


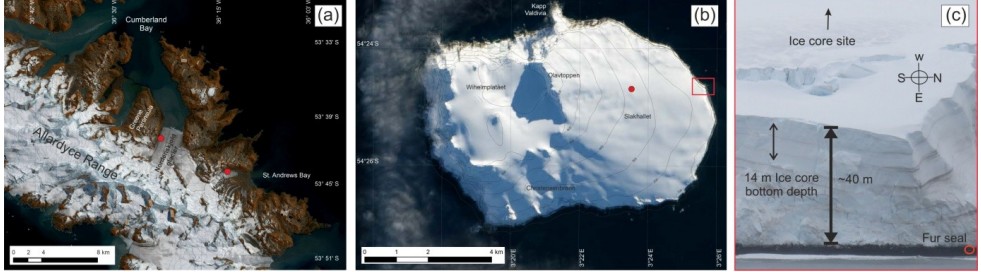

**Figure 4: Atlantic sector ice cores. (a) Map of central South Georgia, showing the location of the two ice cores from the Nordenskjold Glacier and Heany Glacier. (b) Map of Bouvet Island, showing ice core location (red circles) and (c) picture of Bouvet ice cliff from area marked by red box. Image a, landsat image courtesy of South Georgia GIS, hosted by the British Antarctic Survey, and image b, credit Google Earth. Contours derived from TanDEM-X 90m DEM**


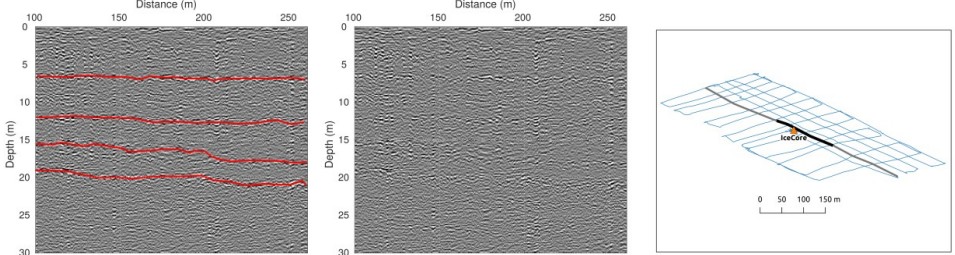

**Figure 5: Radargram representing a section of a transect crossing the ice core extraction point at Mertz 1. The radargram is shown with detected layer in red (left) and without layer interpretation (center). X-axis shows the distance covered within the transect. Y-axis represents the estimated depth. The reference map (right) depicts the full track of measurements (blue), the profile (grey) and the section shown in the radargram (black).**





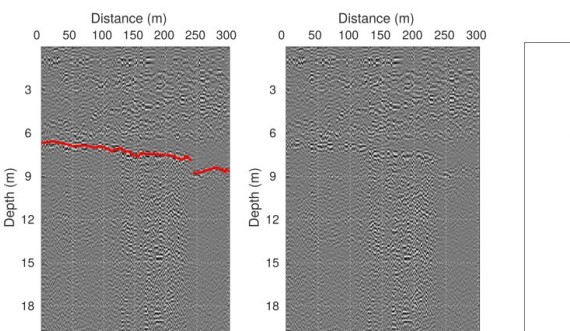
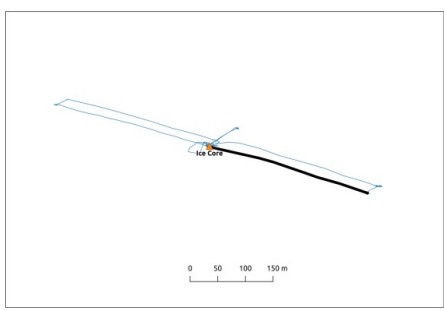

**Figure 6: Radargram collected at the 'fast ice' in Mertz. Profile obtained starting at the ice-core position. The radargram is shown with detected layer in red (left) and without layer interpretation (center). X-axis shows the distance covered within the transect. Y-axis represents the estimated depth. The reference map (right) depicts the full track of measurements (blue), the profile (grey) and the section shown in the radargram (black).**



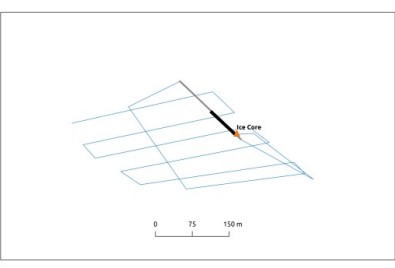

**Figure 7: Section of a radargram obtained walking in direction NW from the ice core position at Young Island. A crevasse was detected at 40 m from the starting point (discontinuity in the red picked layer). The radargram is shown with detected layer in red (left) and without layer interpretation (center). X-axis shows the distance covered**
**within the transect. Y-axis represents the estimated depth. The reference map (right) depicts the full track of measurements (blue), the profile (grey) and the section shown in the radargram (black).**

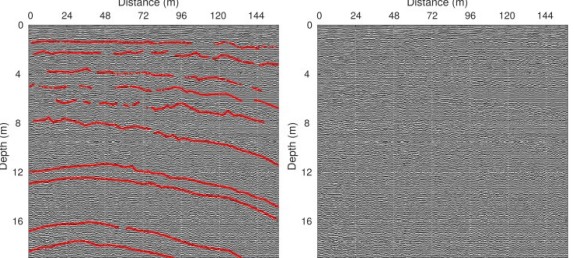
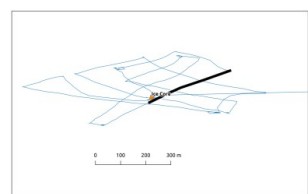




**Figure 8: Radargram obtained at Mount Siple ending at the position of the ice core extraction. The radargram is shown with detected layer in red (left) and without layer interpretation (center). X-axis shows the distance covered within the transect. Y-axis represents the estimated depth. The reference map (right) depicts the full track of measurements (blue), the profile (grey) and the section shown in the radargram (black).**


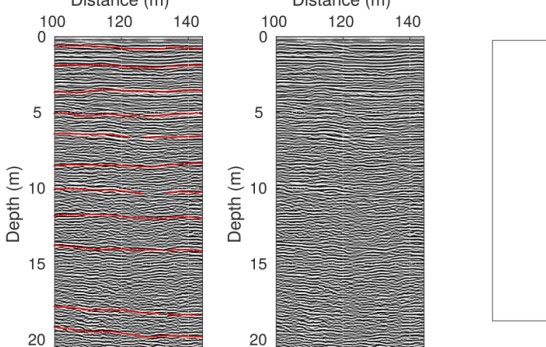
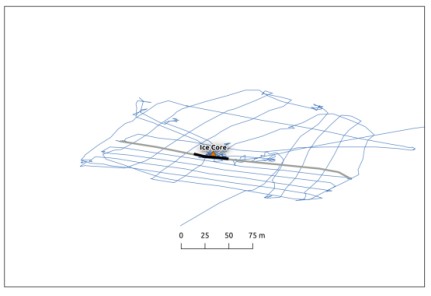

**Figure 9: Section of a radargram taken in a east-westerly direction at Peter 1[st]. The radargram is shown with detected layer in red (left) and without layer interpretation (center). X-axis shows the distance covered within the transect. Y-axis represents the estimated depth. The reference map (right) depicts the full track of measurements (blue), the profile (grey) and the section shown in the radargram (black).**


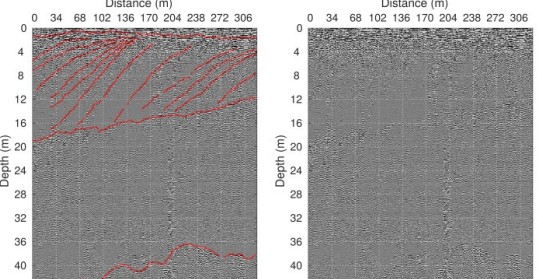
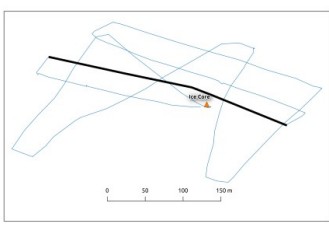


**Figure 10: Radargram obtained at Bouvet going downward (direction SE) from 346 to 329 m a.s.l. The radargram is shown with detected layer in red (left) and without layer interpretation (center). X-axis shows the distance covered within the transect. Y-axis represents the estimated depth. The reference map (right) depicts the full track of measurements (blue), the profile (grey) and the section shown in the radargram (black).**


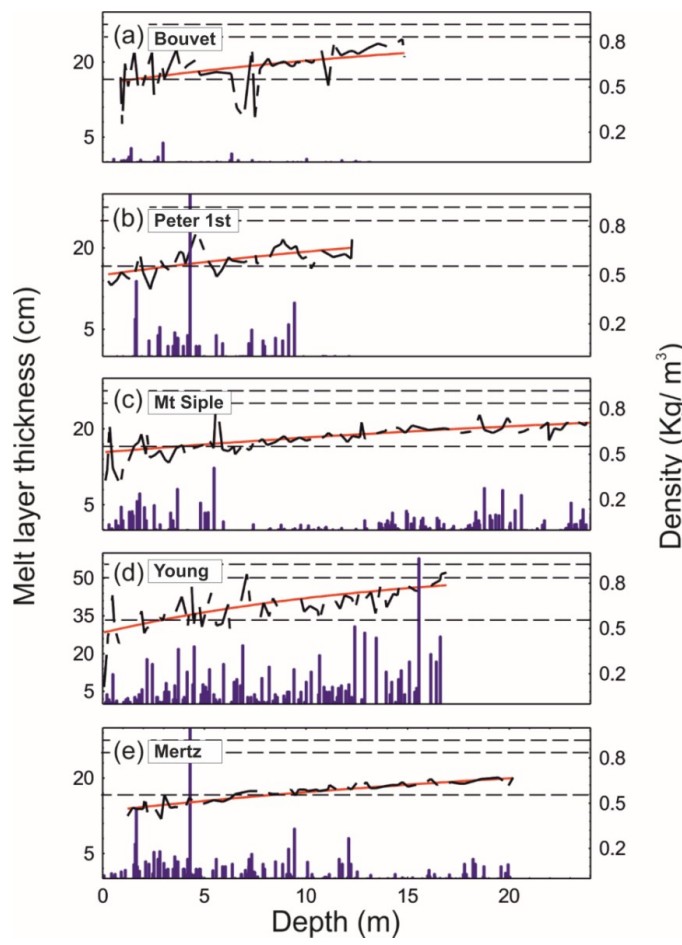

**Figure 11: Melt layer thickness (blue/ cm), measured snow density (black) and fitted density curve (red) for each site (a) Bouvet, (b) Peter 1st island, (c) Mt Siple, (d) Young Island and (e) Mertz 1. Note the axis change for plot d (Young Island). The horizontal lines represent the critical density (0.55 Kg m-3), the pore close-off depth (0.83 Kg m-3) and ice density (0.917 kg m-3).**






| Site name | Latitude | Longitude | Elevation (m) | Depth (m) | ERA-5 average monthly (max) temperature (°C), 1979-2017. | | AWS max recorded temperature (°C) during years of operation. | |
|---|---|---|---|---|---|---|---|---|
| | | | | | 2-m | Elevation corrected | AWS | Elevation corrected |
| Mertz | 67° 33' 34" S | 145° 18' 45"E | 320 | 20.48 | -15.24 (-4.08) | -17.40 (-6.26) | N/A | N/A |
| Mertz fast ice | 67° 26' 28" S | 145° 34' 28"E | 6 | 9.31 | | -15.26 (-6.28) | N/A | N/A |
| Young | 66° 31' 44" S | 162 33' 21"E | 238 (30) | 16.92 | -7.58 (0.63) | -9.19 (-0.98) | 4.2 (1991-1997) | 2.78 |
| Mt Siple | 73° 19' 14" S | 126° 39' 47" W | 685 (230) | 24.14 | -9.58 (-0.29) | -14.24 (-4.95) | 5.5 (1992-2015) | 2.40 |
| Peter 1st | 68° 51' 05" S | 90° 30' 35" W | 730 (90) | 12.29 | -4.56 (1.25) | -9.51 (-3.71) | N/A | N/A |
| South Georgia – Nordenskjold | 54° 22' 57" S | 36° 23' 98" W | 50 (4) | 2.20 | 1.24 (9.68) | 0.9 (9.34) | 8.4 (1903-2018, Grytviken/ King Edward Point) | 8.06 |
| South Georgia - Heany | 54° 25' 91" S | 36° 13' 46" W | 50 (4) | 2.36 | | | | |
| Bouvet | 54° 25' 19" S | 03° 23' 27" E | 350 (42) | 14.07 | -0.5 (3.3) | -2.88 (0.92) | 8.8 (1997-2005) | 6.70 |

**Table 1: Site meta-data for the subICE cores. Measured GPS location, elevation of ice core (with elevation of AWS shown in brackets) and ice core borehole depth. Annual average temperature from ERA-5 (1979-2017) at 2-m elevation and corrected for ice core elevation (using a lapse rate of 0.68 deg / 100 m), with the maximum monthly temperature shown in brackets. Where available the maximum recorded AWS temperature is shown, at the AWS elevation and corrected for ice core elevations, with years of operation in brackets.**




| Site Name | Date | Wave velocity (m/ns) | Max. depth (m) of layer detected | Description | Melt frequency (layer m$^{-1}$) and percentage | Average thickness (cm) | Maximum thickness (cm) |
|---|---|---|---|---|---|---|---|
| Mertz 1 | 29/01/17 | 0.204 | 63 | Relatively flat layering. 13–14 layers detected. There is one clearer at ~20m | 5.26 (6%) | 0.30 | 3.98 |
| Fast Ice Mertz | 30/01/17 | 0.206 | 7.6 | Only one layer detected at 7.6 m (average depth) | 6.02 (3%) | 1.83 | 30 |
| Young | 4/02/17 | 0.2 | 36 | Highly compacted layering. Multiple crevasses. Layers are visible but they are close to each other and merge. | 7.66 (47%) | 1.32 | 12.5 |
| Mount Siple | 11/02/17 | 0.204 | 36 | Clear layering. > 10 traceable layers Smooth changes in the ice structure. | 7.45 (10%) | 6.57 | 61 |
| Bouvet | 12/03/17 | 0.194 | 41 | Highly variable snow layers. Consistent reflector at ~40m. | 6.45 (16%) | 0.69 | 3 |
| Peter 1st | 15/02/17 | 0.202 | 43 | Relatively continuous layering. 15 traceable layers. Smooth changes in the ice structure. Presence of crevasses | 4.69 (11%) | 1.03 | 8.1 |

**Table 2: General description of the subsurface for each site based on the visualisation of the radargrams. Frequency, percentage (% of whole core) and thickness of observed melt layers, corrected for thinning. Melt layers in the Mertz**

**fast ice core were only measured in the top three bags (0-2.4 m).**







| Site name | Zero-depth intersection (m) | Close-off depth (m) | Estimated ice core bottom age (AD) | Annual average P-E (1979-2017) | Deepest GPR layer detected (m) | Estimated age at deepest observed GPR depth (AD) | Estimated age at 100 m depth (AD) |
|---|---|---|---|---|---|---|---|
| Bouvet | 0.523 | 28.65 | 2001 (+/- 2) | 0.59 (0.59) | 41 (bedrock) | 1962 (+/- 5) | N/A |
| Peter 1st | 0.500 | 34.52 | 2001 (+/- 2) | 0.44 | 25 | 1948 (+/- 5) | 1836 (+/- 10) |
| Mt Siple | 0.512 | 51.90 | 1997 (+/- 3) | 0.86 | 30 | 1985 (+/- 5) | 1851 (+/- 10) |
| Young | 0.472 | 21.09 (16.62) | 1997 (+/- 3) | 0.58 | 34 | 1967 (+/- 5) | 1743 (+/- 10) |
| Mertz 1 | 0.446 | 50.48 | 1992 (+/- 3) | 0.48 | 63 | 1919 (+/- 5) | 1742 (+/- 10) |
| Mertz fast ice | 0.031 | 7.05 | 2010 (+/- 1) | 0.48 | 6.2 | 2010 (+/- 5) | N/A |

**Table 3: Densification at each site. Values derived from the fitted density equation to estimate zero-depth intersection and close-off depth (measured values shown in brackets). Annual average P-E from ERA-5 (1979-2017) compared with annual layer counted estimate (meters water equivalent) for Bouvet shown in brackets (King et al., 2019).**

**Estimated ages at ice core bottom depths, depths of deepest GPR layer and depth at 100 m, based on the fitted density profiles.**