# Peer review of "Physical properties of shallow ice cores from Antarctic and sub-Antarctic islands"

_The Cryosphere, 2020_

## Referee Comment (RC1) · Howard Conway (Referee) · 30 Jun 2020

ORIGINALITY

The manuscript reports glaciological field studies conducted as a part of the Antarctic Circumnavigation Expedition during the 2017 austral summer. Here the authors present depth profiles of density and melt layers from short (14-24m) firn cores extracted from three sub-Antarctic islands and two coastal domes. Short radar surveys in the vicinity of the cores are also presented. The study is a reconnaissance to evaluate potential ice-core sites that preserve records of past climate and atmospheric circulation in this important, data-sparse region.

SCIENTIFIC QUALITY

[Figure]

Concerning eqn.1, does ph denote density at depth h? Is model parameter "a" equivalent to the "Zero-depth intersection (m)" given in Table 3? Is model parameter "b" also given in Table 3?

How does the presence of melt layers affect the Herron and Langway age-density model? How do you estimate the age uncertainties given in Table 3? How might annual or decadal variations in accumulation impact the age-scale? A more rigorous analysis of uncertainties is needed.

Detailed measurements of stable isotopes, ions, and organic chemistry from the Bouvet Island core have been previously reported and interpreted (King et al., 2019). The 14.2m core was dated using annual cycles in deuterium and calcium. Assuming data shown in Fig. 11 are correct (see comments below about inconsistencies), the core from Bouvet Island appears to be the least affected by melt layers. However, Fig. S1 in King et al. (2019) shows the isotope and chemistry signals are strongly attenuated near the bottom of the core.

I see (line 309) that annual layer counting of the other SUBICE cores has not yet been completed. Is it in progress? The paper would be much stronger if measurements of isotopes and chemistry and an associated age scale for the other cores are included. An age scale for each core is needed to validate the use of the ERA-5 derived accumulation rates, and to establish the fidelity of the age-scales.

Apart from identifying a possible basal reflection at Bouvet Island, it is not clear how the discussion of the radar profiles support the focus of this study.

At 400MHz reflections are more sensitive to changes in density than changes in chemistry. In this case, one might expect that shallow radar reflections (where the background density is less that $\sim700$kg/m3) might correspond to melt layers. However as presented, it is difficult to determine if there is such a correspondence. As mentioned in the text, the radar system records the two-way travel time (TWT time) to a reflector. For matching radar layers with the core stratigraphy, rather than using an average wave

speed (as implied in Table 2) it would be best to use an appropriate depth profile of the dielectric constant to estimate variations in the wave speed through the firn.

Further, it would be informative to show the location of the core site directly on the radargrams, and to evaluate the mismatch between radar-detected layers and the melt stratigraphy in the cores.

SIGNIFICANCE

This reconnaissance study identifies several potential sub-Antarctic ice-core sites that contain a centennial-scale climate record. All sites contain melt layers, but progress has been made dating cores in sub-polar and maritime climates (e.g Abram et al., 2013; Neff et al 2017). Although perhaps more logistically challenging, it may also be possible to find suitable sites at higher elevations on Bouvet Island, Mount Siple or South Georgia.

PRESENTATION QUALITY

Are columns "melt frequency", "average thickness" and "maximum thickness" given Table 2 derived from "visualization of the radargrams" or are they derived from the cores?

Results shown and discussed in the text, Table 2, and Fig. 11, are inconsistent. For example, Table 2 indicates the thickest ice layer at Mt Siple is 61cm, while Fig. 11 indicates that it is about 12cm, and at Peter 1 Island the thickest layer is 8.1cm, but Fig 11 suggests it is >30cm. Text (line 303ff) states: The average melt layer thickness in the Bouvet core is 0.3 cm, observed at a frequency of five layers per meter; with the largest measured melt layer just 3.98 cm (Table 2)....., in contrast, Table 2 shows values 0.69cm, 6.5 layers/m, and 3cm for the largest melt layer.

Section 2.2.5 It is not clear why the two cores from South Georgia are mentioned here since they ..."do not provide contemporary climate information". However interestingly, Mayewski et al (2016) presented results of an initial reconnaissance for an ice core site

on South Georgia and suggested that annual stratigraphy might be preserved at sites with elevations above 2000m.

---

## Referee Comment (RC2) · Anonymous Referee #2 · 15 Jul 2020

The manuscript by Thomas et al. presents a preliminary analysis of the potential of a number of ice cores from sub-Antarctic islands for short-term climate reconstructions. The data are novel and exciting, though this is somewhat undersold until the Conclusions. The structure of the manuscript could be much improved; in particular, the Methods need to be complete (including uncertainties) and a separate Discussion section would allow for more detailed and clearer interpretations. I also wonder if the title best summarises the manuscript, as the ultimate goal is currently to calculate the bottom age of a longer core at each site (though no climate reconstruction is actually carried out) – certainly to keep the title, more description (and analysis) would be required. There are a large number of minor technical inconsistencies that need correcting.

Major comments

1. The manuscript would benefit from having a Discussion section. At present, the small amount of discussion is mixed in with the Results, and as such, is lacking in detail. I at least expected a discussion of the suitability of each of the sites (ice conditions and location as that was a primary focus in the Methods) – some of this is hidden in the Results and would be much clearer in the Discussion, perhaps organised with a sub-section structure for this section that answers each of the aims.

2. The structure of the Methods and Figures 2-5 could be improved. I would expect the field site/ice core descriptions to come first, followed by ice core analysis, GPR, then met data. This makes more sense with the order the aims are presented in the preceding paragraph (and would make more sense for the structure of the following sections as well). The Methods are incomplete in places and some important details are floating around in the Results/Discussion section rather than the Methods. In addition, I don't see the value in separating Figures 2-5 into sea sectors when this isn't referred to elsewhere and when the Methods describe each site in turn. I suggest either dividing the Method sub-sections by sea sector (if that makes sense with discussion later in the manuscript), or combining these figures into one – perhaps as subpanels on Figure 1, colour/shape-coded by site. It would also be useful to show the GPR lines on these figures as well as the ice core locations.

3. A large part of the Results focuses on the GPR results but is isn't clear what these data ultimately contribute to other than general characterisation and providing ice depth for one site – they are set up as if they will provide much more than this. It almost feels like the data analysis is not complete(ly presented). Layers are identified in the radargrams, but this seems a little haphazard and is not compared with the ice cores – why? 4. Uncertainties for the measured densities and fitted density curve are not presented, and the calculations for the calculated depths are not explained?

Minor comments

L21 and throughout: Please be consistent with hyphenation; e.g. ground-penetrating

radar, pore close-off

L28: Units should be kg m-3?

L33-53: These first four paragraphs are a little repetitive and double back on themselves. I think it could be condensed into two paragraphs, with, for example, the first explaining why climate records are important and missing in this region, and the second detailing the potential of SAIs. This would lead nicely into the few paleoclimate records that do exist.

L57 and throughout: Be consistent in use of comma to denote thousands in numbers

L58: Remove comma after limitation

L71: Terminus should be termini

L74: How much retreat has occurred – how much of the record is lost?

L83: Why the upper 40 m? The longest ice core was 24 m, and the GPR up to 51 m so this seems quite a strange depth to choose. Perhaps instead state the "upper ice column"?

L88: ECMWF acronym needed after this first use

L90 and throughout: Either use "degree" or the symbol, unless there is a reason for swapping between these?

L91: Comma needed before "however"

L95-100: Links and DOIs might be better provided in the reference list rather than in-text. This would allow a clearer description of the Bouvet station, for example (what does WMO stand for?). You state in this paragraph what data is available, but what is actually used for this study?

L97: Unnecessary comma and missing space after "Island"

L107: Use defined SAI acronym
L108-109: This sentence belongs in the Results

L150: Missing space between 3,110 and m

L152 and throughout: "average annual temperatures" should be singular

L153: You haven't defined or used a.s.l. as an abbreviation prior to now – either use throughout or don't

L173-174: The phrasing of this sentence is confusing – if the cores aren't used, then how are the bottom ages obtained? If they aren't used, I simply wouldn't mention them at all (save for your future study), but if they are, perhaps present this site at the end of this sub-section and describe what you do do with the core (differently to the others)

L174: Elevation, not altitude

L178 and throughout: "islands" should be "island's"

L188: I think "extent" should be "extend"

L198: GPR has been used prior to this (without the acronym. . .)

L226: How much correction was applied for ice thinning – is it possible to give an idea with a mean and range?

L226: "that" should be "which"

L228: Where are the methods for estimating the bottom depth of each site? Found – Sections 3.3 and 3.4; I think these would make more sense here at the end of the Methods, with more detail than currently provided (how are uncertainties calculated?)

L236-238: The number of clauses in this sentence makes it hard to follow. Perhaps rewrite to something like: "Layers could not be distinguished in the upper ∼7 m of snow and only reflected weakly beneath this depth; 11 distinct, but discontinuous, layers were identified down to a depth of 62 m. We estimate this to be the..."

L245: Typical of what?

LL268-269: I don't see the multiple layers you interpret in the radargram. Perhaps they would be clearer if you made the radargrams larger or only showed a section of the image currently presented?

L275: Have you only shown some of the layers in Figure 9 – there appear to be many more?

L281: I see the horizontal layers discussed here, but not the nearer-vertical layers that are shown in Figure 10? Is this again a size-resolution issue with the figure?

L303: "effected" should be "affected"

L309: Thus far, the project has been subICE not SUBICE

L317: AWS has already been used so doesn't need defining here

L320: What could have caused the large number of melt layers at Young Island if not surface temperature? Is it possible to plot the number of positive degree days from the AWS (admittedly over a short period)? A discussion of the depth distribution of these layers might be interesting and help to suggest other causes

L356: This sentence is incomplete

L365: If the annual layer counting was done in this study please describe in Methods, or if another please provide a reference

L371: Table 5 should be Table 3?

L412: Estimate should be plural. Is this sentence stating that only Bouvet provided an ice thickness estimate?

L414: I think core should be plural

Figures 5-10: It would make more sense to me to have the reference map first, then the uninterpreted radargram, then the interpreted radargram on the far right. All these figures need panel labels for consistency with other figures, and the captions can be

shortened (e.g. axes do not need to be described). Font size could be larger and some of the radargrams, as previously mentioned. The figures would also be neater if the reference map panel was the same height as the other panels and all the of the tracks filled each box. The mid-panel y-axis on Figure 7 is obscured.

---

## Author Comment (AC1) · 28 Sep 2020

Response to reviewer 1

We thank reviewer Howard Conway for his evaluations and suggested improvements. We have addressed all the concerns and made the suggested revisions to the text and figures. Below we show the reviewer comments in black, and our response in red.

ORIGINALITY
The manuscript reports glaciological field studies conducted as a part of the Antarctic Circumnavigation Expedition during the 2017 austral summer. Here the authors present depth profiles of density and melt layers from short (14-24m) firn cores extracted from three sub-Antarctic islands and two coastal domes. Short radar surveys in the vicinity of the cores are also presented. The study is a reconnaissance to evaluate potential ice-core sites that preserve records of past climate and atmospheric circulation in this important, data-sparse region.

SCIENTIFIC QUALITY
Concerning eqn.1, does ph denote density at depth h? Is model parameter "a" equivalent to the "Zero-depth intersection (m)" given in Table 3? Is model parameter "b" also given in Table 3?

Yes, ph is density at depth h. This has been added to the text. The values of a and b are not currently provided in the table but they could be added.

How does the presence of melt layers affect the Herron and Langway age-density model?

Based on the Herron and Langway densification model the dominant densification process in the upper part ($\rho \leq 550$ kg m$^{-3}$) are grain settling and packing of snow grains (Herron and Langway, 1980). However, undoubtably refreezing of melt water and/or liquid precipitation also cause firn to densify. Thus, our approach is very simplified. It would have been preferable to use an improved model that can account for the influence of melt (eg Lightenberg et al., 2011), however the current observations for these sites do not allow for this.

In a steady-state profile, the meltwater refreezing and densification cannot be modelled, as they move to greater depths with time. If we assume that heat is released upon refreezing, the local firn temperature will increase, further accelerating the densification. At the same time, ice layers lead to a higher average density, thereby reducing the potential densification rate.

Although not ideal, the measured density value used in the equation will reflect the presence of melt. The values of surface snow (or zero-intersection depth) are considerably higher than those modelled or observed for coastal Antarctica.

As discussed below, we have only annual layer counted the Bouvet core to date. However, this allows us to demonstrate that the simple age-density model produced the same bottom age using both methods. Although we appreciate that the Bouvet core was less susceptible to melt than the other island sites.

The scope of this paper was to present the new data collected from these remote islands as a first step to establishing if deeper drilling projects would be feasible. We suggest adding additional caveats and expand the explanations about the influence of melt on the depth-age model used.

How do you estimate the age uncertainties given in Table 3? How might annual or decadal variations in accumulation impact the age-scale? A more rigorous analysis of uncertainties is needed.

Snow accumulation variability at both annual and decadal timescales will undoubtably impact the estimated age-scale. These alterations were considered when estimating the errors presented in table 3, however we agree a more detailed description of how those numbers were derived is required. We note that ERA-5 does not indicate a notable trend in snow accumulation over these timescales, however as we have already established ERA-5 does not have the resolution to capture the islands. Observational records from coastal Antarctic ice cores confirm large trends in snow accumulation are possible, however generally not since 1997 (the estimated length of our records).

The age-density calculations were run using the annual average snow accumulation from ERA5. We added (subtracted) one standard deviation to represent an upper (lower) accumulation estimate. The estimated error in the bottom age is simply the difference between the upper and lower age estimates.

We have updated the text to include a more comprehensive description of error calculations.

Detailed measurements of stable isotopes, ions, and organic chemistry from the Bouvet Island core have been previously reported and interpreted (King et al., 2019). The 14.2m core was dated using annual cycles in deuterium and calcium. Assuming data shown in Fig. 11 are correct (see comments below about inconsistencies), the core from Bouvet Island appears to be the least affected by melt layers. However, Fig. S1 in King et al. (2019) shows the isotope and chemistry signals are strongly attenuated near the bottom of the core. I see (line 309) that annual layer counting of the other SUBICE cores has not yet been completed. Is it in progress? The paper would be much stronger if measurements of isotopes and chemistry and an associated age scale for the other cores are included. An age scale for each core is needed to validate the use of the ERA-5 derived accumulation rates, and to establish the fidelity of the age-scales.

We are not able to present the annual layer counting for all the sites presented. The dating and climatological interpretation is currently being completed as part of multiple PhD projects. As such I would prefer to keep those studies separate. To date, only the Bouvet record has been dated and presented in King et al., 2019.

Apart from identifying a possible basal reflection at Bouvet Island, it is not clear how the discussion of the radar profiles support the focus of this study. At 400MHz reflections are more sensitive to changes in density than changes in chemistry. In this case, one might expect that shallow radar reflections (where the background density is less that ~700kg/m3) might correspond to melt layers. However as presented, it is difficult to determine if there is such a correspondence. As mentioned in the text, the radar system records the two-way travel time (TWT time) to a reflector. For matching radar layers with the core stratigraphy, rather than using an average wave speed (as implied in Table 2) it would be best to use an appropriate depth profile of the dielectric constant to estimate variations in the wave speed through the firn. Further, it would be informative to show the location of the core site directly on the radargrams, and to evaluate the mismatch between radar-detected layers and the melt stratigraphy in the cores.

The reviewer has correctly pointed out that in our case the reflections are given by density changes. The aim of the GPR data in this paper was a description of the observations made and a preliminary analysis to characterize each ice core site. For this purpose, as we stated in Line 220, we used the density profile to estimate the mean velocity propagation for each site.

We initially omitted to perform a detailed analysis of the detected layers depth given the propagation velocity variations through the firn. However, we have now performed a (graphical) comparison of layers detected in the GPR data with the corrected depth estimation based on the variability of the velocity propagation, and the melt layers from the ice core.

The analyses had the following steps:

1. Using the density data to get a propagation velocity/depth profile.

2. Manually picking of layers in the radargram section close to the ice core. Estimating the layers depth according to the velocity for each layer.

3. Using melt layers from the ice core data to compare with the depth layer interpretation.

Result:

We assumed a range of visibility of 35 cm for the GPR (theoretical resolution) and counted the amount and thickness of the melt layers observed in the ice core within this range. We analyzed how strong the reflector would be given the amount of ice and plot the depth points that would be strong enough to be detected by the GPR. The figure shows the potentially visible patterns of ice for the GPR (blue dots) and the depth of the layers interpreted in the radargram (red dots). The layers continuity has been assessed in profile sections of 20 m length near the ice core extraction and the mean depth layers has been used (±std in small red dots).

The GPR detection is subject to several sources of uncertainty. Two of which are particularly decisive in the error: manual picking of the layers in the radargram and the distance between the ice core extraction and the actual radargram profile.

[Figure]

The comparison is very poor particularly for Young Glacier, which ice core had a high amount of thin ice lenses. The radargrams for this site show unclear continuity of the layering and merging of single reflectors.

Some sites show layers detection shifted in depth. This can be due to the spatial variation of the layer depth, given that the radargrams were taken in the surrounding area of the ice core extraction (separation 10-20m),

Other differences can be attributed to the fact that some ice lenses are local and although an ice lens has been detected in the ice core, it does not correspond to a continuous layer through the surrounding area.

SIGNIFICANCE This reconnaissance study identifies several potential sub-Antarctic ice-core sites that contain a centennial-scale climate record. All sites contain melt layers, but progress has been made dating cores in sub-polar and maritime climates (e.g Abram et al., 2013; Neff et al 2017). Although perhaps more logistically challenging, it may also be possible to find suitable sites at higher elevations on Bouvet Island, Mount Siple or South Georgia.

PRESENTATION QUALITY
Are columns "melt frequency", "average thickness" and "maximum thickness" given Table 2 derived from "visualization of the radargrams" or are they derived from the cores?

This is from visual interpretation from the ice cores. We will make this clearer in the text.

Results shown and discussed in the text, Table 2, and Fig. 11, are inconsistent. For example, Table 2 indicates the thickest ice layer at Mt Siple is 61cm, while Fig. 11 indicates that it is about 12cm, and at Peter 1 Island the thickest layer is 8.1cm, but Fig 11 suggests it is >30cm. Text (line 303ff) states: The average melt layer thickness in the Bouvet core is 0.3 cm, observed at a frequency of five layers per meter; with the largest measured melt layer just 3.98 cm (Table 2)....., in contrast, Table 2 shows values 0.69cm, 6.5 layers/m, and 3cm for the largest melt layer.

Apologies, the site names were incorrect in the table. Table 2 has been updated with the correct values to match figure 11.

Section 2.2.5 It is not clear why the two cores from South Georgia are mentioned here since they ..."do not provide contemporary climate information". However interestingly, Mayewski et al(2016) presented results of an initial reconnaissance for an ice core site on South Georgia and suggested that annual stratigraphy might be preserved at sites with elevations above 2000m.

Agreed. Reference to the South Georgia cores has been removed.

---

## Author Comment (AC2) · 28 Sep 2020

Response to reviewer 2

We thank reviewer 2 for their evaluations and suggested improvements. We have addressed all the concerns and made the suggested revisions to the text and figures. Below we show the reviewer comments in black, and our response in red.

The manuscript by Thomas et al. presents a preliminary analysis of the potential of a number of ice cores from sub-Antarctic islands for short-term climate reconstructions. The data are novel and exciting, though this is somewhat undersold until the Conclusions.

The structure of the manuscript could be much improved; in particular, the Methods need to be complete (including uncertainties) and a separate Discussion section would allow for more detailed and clearer interpretations.

Thank you for your suggestions. We agree that the conclusions were a little undersold and that expanding on the methods and discussion sections will improve this.

I also wonder if the title best summarises the manuscript, as the ultimate goal is currently to calculate the bottom age of a longer core at each site (though no climate reconstruction is actually carried out) – certainly to keep the title, more description (and analysis) would be required.

We would prefer to keep the title for consistency. We hope that the expanding discussion section (see below) and enhanced description of the GPR reflections and how they relate to the ice core will allow this. At this stage we are not presenting any climate reconstructions.

There are a large number of minor technical inconsistencies that need correcting. Major comments

1. The manuscript would benefit from having a Discussion section. At present, the small amount of discussion is mixed in with the Results, and as such, is lacking in detail. I at least expected a discussion of the suitability of each of the sites (ice conditions and location as that was a primary focus in the Methods) – some of this is hidden in the Results and would be much clearer in the Discussion, perhaps organised with a subsection structure for this section that answers each of the aims.

Agreed. We have now separated the results and discussion section to include sub-heading addressing the main issues as expressed in the introduction. 1) evaluate the ice conditions and internal layering in the upper ice column, 2) determine the extent of surface melt and 3) estimate potential bottom ages of future deep drilling expeditions.

2. The structure of the Methods and Figures 2-5 could be improved. I would expect the field site/ice core descriptions to come first, followed by ice core analysis, GPR, then met data. This makes more sense with the order the aims are presented in the preceding paragraph (and would make more sense for the structure of the following sections as well).

Agreed. The methods sections have been restructured.

The Methods are incomplete in places and some important details are floating around in the Results/Discussion section rather than the Methods. In addition, I don't see the value in separating Figures 2-5 into sea sectors when this isn't referred to elsewhere and when the Methods describe each site in turn. I suggest either dividing the Method sub-sections by sea sector (if that makes sense with discussion later in the manuscript), or combining these figures into one – perhaps as subpanels on Figure 1, colour/shape-coded by site. It would also be useful to show the GPR lines on these figures as well as the ice core locations.

The intention was to avoid a large and over-crowded figure, hence separating into smaller sections. However, we can revise this and put all the location maps into one figure and remove the reference to sea sectors. In combining the figures in this way, it might be too small to see the GPR lines but we can add those too.

3. A large part of the Results focuses on the GPR results but is isn't clear what these data ultimately contribute to other than general characterisation and providing ice depth for one site – they are set up as if they will provide much more than this. It almost feels like the data analysis is not complete(ly presented). Layers are identified in the radargrams, but this seems a little haphazard and is not compared with the ice cores – why?

Our intention was never to directly compare the ice core observations with the layer detection of the GPR. The resolution of the two methods is very different. The ice core shows ice lenses in mm, while the GPR layers have a resolution of about 30 cm. Thus, a single thin layer (<30 cm) would not be seen in the radargram.

The value of the GPR measurements is that it helps to characterise the site over a larger area and investigate the spatial distribution eg stratigraphy, continuity and number of layers. It is also important to establish the likelihood that a melting event observed in the ice core is a well distributed layer or a localised event. We have now expanded the GPR description (see comments to reviewer 1) to include a detailed comparison of the GPR layers with the observed melt layers in the ice core.

4. Uncertainties for the measured densities and fitted density curve are not presented, and the calculations for the calculated depths are not explained?

Agreed. Additional descriptions of the depth uncertainties have been included and estimated uncertainties added.

Minor comments

L21 and throughout: Please be consistent with hyphenation; e.g. ground-penetrating radar, pore close-off

Revised.

L28: Units should be kg m-3?

Revised.

L33-53: These first four paragraphs are a little repetitive and double back on themselves. I think it could be condensed into two paragraphs, with, for example, the first explaining why climate records are important and missing in this region, and the second detailing the potential of SAIs. This would lead nicely into the few paleoclimate records that do exist.

The introduction has been re-structured and condensed.

L57 and throughout: Be consistent in use of comma to denote thousands in numbers

Revised.

L58: Remove comma after limitation

Revised.

L71: Terminus should be termini

Revised.

L74: How much retreat has occurred – how much of the record is lost?

The referenced paper describes 97% retreat of the island's glaciers. However, this occurs at the margins and it is difficult to determine how much of the paleoclimate record has been lost.

L83: Why the upper 40 m? The longest ice core was 24 m, and the GPR up to 51 m so this seems quite a strange depth to choose. Perhaps instead state the "upper ice column"?

We agree, upper ice column is more appropriate. The maximum expected ice depth was ~25 m, therefore we choose a frequency to best capture this.

L88: ECMWF acronym needed after this first use L90 and throughout: Either use "degree" or the symbol, unless there is a reason for swapping between these?

Revised.

L91: Comma needed before "however"

Revised.

L95-100: Links and DOIs might be better provided in the reference list rather than intext. This would allow a clearer description of the Bouvet station, for example (what does WMO stand for?). You state in this paragraph what data is available, but what is actually used for this study?

The text has been condensed and DOIs added to reference list. The data provide estimates of site temperatures, which is used in the discussion regarding surface melt.

L97: Unnecessary comma and missing space after "Island"

Revised.

L107: Use defined SAI acronym

Revised.

L108-109: This sentence belongs in the Results

Revised.

L150: Missing space between 3,110 and m

Revised.

L152 and throughout: "average annual temperatures" should be singular

Revised.

L153: You haven't defined or used a.s.l. as an abbreviation prior to now – either use throughout or don't

Revised.

L173-174: The phrasing of this sentence is confusing – if the cores aren't used, then how are the bottom ages obtained? If they aren't used, I simply wouldn't mention them at all (save for your

future study), but if they are, perhaps present this site at the end of this sub-section and describe what you do do with the core (differently to the others)

Agreed. Reference to these sites should be removed.

L174: Elevation, not altitude

Revised.

L178 and throughout: "islands" should be "island's"

Revised.

L188: I think "extent" should be "extend"

Revised.

L198: GPR has been used prior to this (without the acronym...)

Revised.

L226: How much correction was applied for ice thinning – is it possible to give an idea with a mean and range?

We have decided to remove the correction for thinning in the paper because the estimation that thinning is proportional to burial may not be appropriate. More information is needed to establish the rate of thinning in the firn column at sites susceptible to melt and thus we think it better to present the raw (un-corrected) thicknesses in the revised text.

L226: "that" should be "which"

Revised.

L228: Where are the methods for estimating the bottom depth of each site? Found – Sections 3.3 and 3.4; I think these would make more sense here at the end of the Methods, with more detail than currently provided (how are uncertainties calculated?)

Agreed, we will move to methods and expand.

L236-238: The number of clauses in this sentence makes it hard to follow. Perhaps rewrite to something like: "Layers could not be distinguished in the upper~7m of snow and only reflected weakly beneath this depth; 11 distinct, but discontinuous, layers were identified down to a depth of 62 m. We estimate this to be the..."

Sentence updated.

L245: Typical of what?

Added – "typical of surface snow".

L268-269: I don't see the multiple layers you interpret in the radargram. Perhaps they would be clearer if you made the radargrams larger or only showed a section of the image currently presented?

This is because of the resolution of the image. We wanted to show that the layers follow the surface slope, but it will be best to change the figure to show a small section. The figure will be expanded, and a small section will be shown.

L275: Have you only shown some of the layers in Figure 9 – there appear to be many more?

Yes, we have only selected the strongest for this section and those that are the most continuous through the full profile.

L281: I see the horizontal layers discussed here, but not the nearer-vertical layers that are shown in Figure 10? Is this again a size-resolution issue with the figure?

We believe this may be a resolution issue. We are showing the full profile but will update the figure with just the section of the profile to improve this.

L303: "effected" should be "affected"

Revised.

L309: Thus far, the project has been subICE not SUBICE

Revised.

L317: AWS has already been used so doesn't need defining here

Revised.

L320: What could have caused the large number of melt layers at Young Island if not surface temperature? Is it possible to plot the number of positive degree days from the AWS (admittedly over a short period)? A discussion of the depth distribution of these layers might be interesting and help to suggest other causes

This section has been expanded to address this question. Reflecting on the observed increased melt layer thickness with depth, however we include the caveat that the current method of visual layer counting is not sufficient to determine if the layers are a result of an individual melt event or formed from a sequence of smaller events. The aim of this study was to establish if melt was evident at the sites, as a first approach to establishing the islands suitability for deeper drilling. A detailed evaluation of the ice microstructure is currently being conducted as part of a PhD project and we hope to be able to address this issue more thoroughly in future publications.

We evaluate the positive degree days from the AWS. This reveals exceptionally large variability in the 3-hourly data, especially during the winter. We will include more detailed explanation of the AWS data in the revised discussion section.

L356: This sentence is incomplete

Revised.

L365: If the annual layer counting was done in this study please describe in Methods, or if another please provide a reference

Annual layer counting has not been completed for this study. The Bouvet record has been dated and the methods and figures are presented in King et al., 2019.

L371: Table 5 should be Table 3?

Updated.

L412: Estimate should be plural. Is this sentence stating that only Bouvet provided an ice thickness estimate?

Updated and yes, only Bouvet has a clear bed reflection.

L414: I think core should be plural Figures 5-10 - Revised.

It would make more sense to me to have the reference map first, then the uninterpreted radargram, then the interpreted radargram on the far right. All these figures need panel labels for consistency with other figures, and the captions can be shortened(e.g. axes do not need to be described). Font size could be larger and some of the radargrams, as previously mentioned.

Figures update and font increased.

The figures would also be neater if the reference map panel was the same height as the other panels and all the of the tracks filled each box. The mid-panel y-axis on Figure 7 is obscured.

Agreed and revised.

---

## Author Response (AR1)

[revised manuscript text omitted]

Although the resolution of the equipment used allows only a rough comparison with the distribution of melt layers in the ice core, this dataset provides a fair understanding of the spatial distribution of the snow accumulation of each site. Furthermore, we analysed the difference between the layers detected in the radargrams and the melt stratigraphy in the ice core. In order to make the comparison we used a theoretical approach assuming a range of visibility of 35 cm (theoretical resolution) for the GPR. We counted the amount and thickness of the melt layers in the ice core for a moving window in depth. We analysed how strong the reflector would be given the amount of ice and selected the depth points that would be strong enough to be detected by the GPR. The layers continuity of the GPR data have been assessed in profile sections of 20 m length near the ice core extraction (plot c in Figures 2-6) and their mean depth have been used. Subsequently, we compared the potentially visible patterns from the ice core to the depth of the hand picked layers in the radargrams (Figure X).

(Caption) Figure X. Comparison of detected melt layers (horizons) GPR data (red dots, standard deviation in small red dots) and the potentially visible patterns of melt from ice cores for each site.

The presented comparison method is subject to several sources of uncertainty, two of which are particularly decisive: the manual picking of the layer interpretation of the GPR data and the distance between the ice core extraction point and the actual section of the radar data used, which varied from 5-20 m for the different sites. Plus the fact that the average depth in sections of 20 m distance covered was used that added extra uncertainty for the sites which surface and internal stratigraphy were less flat (i.e. Bouvet and Mount Siple).

The relationship is particularly poor for Young Island site which ice core shows a high amount of thin melt layers seen in the radar data as very dense non-continuous layering. Another factor that hinders the matching in the detection is that some melt layers in the ice cores may be local and do not represent a spatially distributed layer.

[revised manuscript text omitted]

---

## Referee Report (RR1)

[referee-annotated manuscript omitted]

---

## Author Response (AR2)

Response to reviewers.

Reviewer 1

Reviewer comment in black and our response in red.

The manuscript by Thomas et al. is much improved on the previous version and was a very enjoyable read. I have just a handful of minor comments to finesse the structure, flow and readability of figures, after which I would recommend it for publication.

Thank you, I agree the paper is much improved and appreciate your input.

L2 and throughout: Unsure why there is now an apostrophe in "island's"? I don't think this is needed in almost all cases.

Checked and updated.

L22: "data were collected"

Corrected.

L69-70: one terminus or several termini? I'm also unclear about the phrase "ice ages of between…" – are you referring to the age of the ice (if so, how from plateaus?) or perhaps the last glacial maximum?

Termini. Sentence updated to clarify "The estimated age of the ice at bedrock for the glacial termini sites is between 8,000-12,000 years old"

L121: 3,110 m above sea level?

Corrected – all updated to m.a.s.l after line 116

L185: "data are"

Corrected

L192-194: Any references to support these values and your particular choice?

Reference Martin and Peel

L207: "ice core extraction site"?

Corrected

L210: Bedrock description is repetitive of L205

Removed repetition.

L219: "were these" should be deleted, I think

Removed.

L246: "altitude corrected" should be "elevation-corrected" – particularly as you use elevation in all later instances

Corrected.

L271: "melt observations … have"

Corrected.

L276: How do you determine which sections would have a strong enough reflection? In addition, while I see why you have included this and the next (misnumbered) section here, it would make more sense to put the methodological parts into the Methods, and present the actual results here.

Corrected numbering.

Reflected layers were picked manually as described in Line 169. "strong enough reflection" is assumed as visible in the profiles at full resolution.

Added extra definition in brackets.

L277: Figure 7 has not yet been mentioned? I assume it is coming later, so these two figures should be swapped in order

Order swapped.

L294-317: Most of this section is interpretations, and should thus be in the Discussion section

Added a sentence referencing the values in table 3 to this section and moved the rest to discussion. Included under an additional subheading 4.1.1

L365: "sites" needs an apostrophe

Corrected.

L392-404: Suggest merging these paragraphs, as the second continues on from the first

Corrected.

L450-452: This paragraph either needs some context/discussion, or to be moved to the Results

Moved to bottom of section 3.5

Figure 1: This is much easier to visualise now, thank you. Just one comment that the scale bars (and coordinates on some panels?) are still too small to read – perhaps pull them out and have them underneath each panel so that they can be larger but not obscure any of the images. The placename fonts on each of these panels also needs to be large enough to read – one way to do this would be to make all the panels much larger.

Figure updated with increased font size and larger scale bars.

Figures 2-6: Thanks again – these are also much easier to interpret. The resolution is a little fuzzy on my document – but this could be the download, and it doesn't preclude ability to see layers you have depicted. Just querying why the picked lines are black in all except Figure 4? Either is fine, but continuity between similar figures is always nice. The panels in Figure 6 would be easier to read if they were a similar size to the previous figure panels – I can't see/read much at present. Panel 6f is not mentioned in the caption, and I wonder if e and f should be swapped around?

Figure 4 and captions updated.

Reviewer 2

The revised manuscript has addressed my earlier concerns - nice work!

Thank you.

I have some minor editorial suggestions for the revised version. In particular:

- the use of apostrophes in the revised text (e.g Island's - lines 2, 18, 19, 459, and more should not have a apostrophe)

Corrected. Overuse of find and replace option in word.

- perhaps "firn cores", or "cores" would be more appropriate than "ice cores"?

OK, I have changed to firn cores throughout the text.

Additional suggestions for your consideration are included in the attached file.

All text changes updated.

And a few other suggestions annotated on the attached revision

Line 271: changed "while the GPR has an expected range of visibility of ~35 cm (theoretical resolution)" to "while the 400 MHz antenna provides a range of visibility of ~35 cm for the dielectric permittivity values at these firn conditions (Koppenjan, 2009; Arcone, 2009).

In addition to the reviewer suggestions we have also added a new reference in the introduction.

Perren, B.B., Hodgson, D.A., Roberts, S.J. et al. Southward migration of the Southern Hemisphere westerly winds corresponds with warming climate over centennial timescales. Commun Earth Environ 1, 58 (2020). https://doi.org/10.1038/s43247-020-00059-6